# A Comparative Study on the Ethical Responsibilities of Key Role Players in Software Development

**Senyeki Milton Marebane [1,\*] and Robert Toyo Hans [2]**

[1] Faculty of Information and Communication Technology, Tshwane University of Technology, Emalahleni 1055, South Africa

[2] Computer Science Department, Tshwane University of Technology, Soshanguve 0152, South Africa; hansr@tut.ac.za

\* Correspondence: marebanesm@tut.ac.za; Tel.: +271-2382-3136

**Abstract:** Background: Issues of lack of consideration for professional responsibility by software engineers (SEs) present major challenges and concerns to software users. Previous studies on the subject of ethical responsibility in software development assessed whether software development key stakeholders should take ethical responsibility for their actions in software development. However, such studies focused on assessing responses from a particular grouping in software development. Objective: Based on the revelation, this study seeks to evaluate the perceived ethical responsibilities in software development by juxtaposing the perceptions of students, educators and industry-based software practitioners on the ethical responsibility of software development key stakeholders in South Africa. Methods: To meet this objective, the study collected data using a survey, which was shared on an online platform. A total of 561 (44 from computing academics; 103 from industry-based software practitioners and 414 from software development students) responses were received. The collected data were analysed using descriptive and variance statistical analysis approaches. Results: The study found that there is no significant statistical difference in how students, educators and software practitioners perceive the ethical responsibility of software development key stakeholders. Conclusions: This finding of the study shows that the prevailing view is that various software development key stakeholders should be held ethically responsible for their contribution to software development. Furthermore, the organisation of ethical responsibilities used in this study provides a useful framework to guide future studies on this subject.

**Keywords:** ethical responsibility; software engineers; students; educators; software practitioners; comparative study

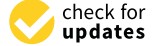



## 1. Introduction

The issue of professional responsibility in the development of information and communication technologies is one of the major concerns in the field [1]. The continued reporting of ethical mishaps resulting from software suggests that software engineers may not be considering their professional responsibility for their work, such as their especial responsibility pertaining to ethical obligations. Being responsible professionally requires practitioners to observe the standards of practice of the profession, assume responsibility for the consequences of work outcomes, behave ethically and safeguard professional and personal integrity. Indicators of one being a responsible professional include taking responsibility for one's work, demonstrating ethical competence, exercising science informed decision-making and adhering to standards of practice. With this said, this article considers and works with the following definition of responsibility given by Solbrekke and Englund [2]: "*a moral obligation assumed by oneself, or bestowed upon a person to be used… to be and act for another*".

Unfortunately, software development work being pressure intensive does not assist the endeavours to promote ethical professional responsibility amongst software developers.

Software developers are inevitably confronted with the constant need to make decisions based on competing needs, such as client demands, costs, operational efficiencies, schedules and technical standards, within limited time frames. These result in time pressures which can adversely affect decision-making [3]. In reviewing previous studies, Kuutila et al. [4] demonstrates how time pressures can also impact negatively on the various aspects of the software process including quality assurance, communication and coordination, software process improvement and user involvement. For example, time pressures lead to compromises on the quality of code and results in costly code reworks. In addition, developers rush to implement solutions or reuse badly written code, resulting in future problems, especially in the mobile development [5] and web development fields [6], which are characterized by limited time frames for innovative products to reach the market. Furthermore, it is also worth noting that the nature of software engineering is complex, imperfect, unpredictable [7] and cannot be changed. It continues to be difficult to foresee and accommodate all unexpected challenges [8]. Regardless of these challenges, Dodig-Crnkovic and Crnkovic [9] affirm that "*It is an engineer's responsibility to foresee and prevent as far as possible any severe consequences of product/system malfunction*". Therefore, software engineers should at all times be alive to their ethical responsibilities.

One of the ways of ensuring that software engineers attend to the above cited challenges, is to bring ethical responsibility to software development practice. For example, the software engineering profession requires its members to practice software engineering in a professional and ethically responsible manner [10]. The realisation of the importance of this has come to the fore for several reasons. Firstly, because software plays an important role in all aspects of humanity; society's increased dependence on technology obviously increases its vulnerability through associated technological failures [11]. Software engineers being conscious about their responsibilities can increase attention towards curbing the vulnerabilities emanating from possible software failures [11]. Secondly, because the software engineering profession advocates for the public good in all its software development initiatives [12], it invokes the need to upscale the assumption of responsibility. Thirdly, because software professionals have an enormous impact on the lives and well-being of others [13], the wielding of power in terms of decision-making and technical expertise inherent in software development requires caution [14]. Furthermore, studies such as those by [6,8,10,15–18] show the need for software engineers' ethical responsibility as part of their professionalism. In the acknowledgement of the importance of taking responsibility in professional practice, it is not surprising that [1] determined that the development of professional responsibility is one of the critical competences focused on in higher education. However, in contrast, [18], in their survey of the literature, found that responsibility as an ethical issue accounted only for 4% of issues identified, the lowest amongst other issues such as agency, autonomy, freedom, identity, justice and privacy. Therefore, attention by researchers on the ethical responsibility to be assumed in software development is necessary.

Despite this heightened need for competencies to assist with responsibility in software development, researchers such as [1,2,16] worryingly cite the lack of focus on professional responsibility. It is therefore important for research studies to investigate if, in practice, there is an assumption of ethical responsibility by those involved in the development of software [1]. Furthermore, considering that software technology evolves, understanding the way practitioners perceive their ethical responsibility provides insights into how they deal with the ethics of newer systems [14]. Therefore, the assessment of whether software development key stakeholders should be held ethically responsible for their contribution to software development is important. However, previous studies about ethical responsibility, although they have studied several aspects of ethical responsibility, have been limited to assessing responses from a singular stakeholder group in software development, such as practitioners alone. As a result, they have lacked focus on comparing responses or perceptions from students learning to become future software engineers, academics contributing to the teaching of software development courses and industry-based software

practitioners. These studies have left a gap in the understanding of the responsibilities of key stakeholders in software development from the perspectives of the above cited software engineers. This study intends to evaluate the perceived ethical responsibilities in software development by juxtaposing the perceptions of students, educators and industry-based software practitioners on the ethical responsibilities of software development key stakeholders. This will assist in bringing to the fore the ethical responsibilities of software development key stakeholders. Based on that, the research question which the study seeks to answer is: "*How do the perceptions of students, educators and industry-based practitioners compare in terms of the ethical responsibility of software development key stakeholders?*".

This study is an extension of a previous study by Marebane et al [17] which will be further discussed in the Section 2. That study was conducted to determine the perceived levels of ethical responsibilities of software engineers, as reported by educators in a South African university. Hence, this new study proceeds to analyse data collected from students, lecturers and software engineers practising in the industry who were asked to respond to a survey about the ethical responsibilities of various software development key stakeholders, such as developers, analysts, quality assurance professionals, management, and users.

The next section presents a literature review outlining the theoretical foundations underpinning this research work. Section 3 outlines the methodology of the study, Section 4 presents and discusses the research results, and Section 5 provides a conclusion and recommendations. Lastly, the limitations of the study and recommendations for future work are outlined in Section 6.

## 2. Literature Review

### 2.1. Ethical Competence of Software Engineers

The complexity of modern-day moral dilemmas associated with advances in computing makes ethical competence an essential skill [19]. This competence can enable software engineers to competently handle ethical challenges in software development. Considering that ethical competence can be developed along with learning about the associated professional responsibility [1,20], the intention of exposing software engineers to ethics is to develop such competence in order to enable them to practically wrestle with the ethical challenges in the development of software.

According to Adam (2006, p. 7) [21] "*Competence' can broadly refer to aptitude, proficiency, capability, skills and understanding*". In simple terms, competence is a combination of knowledge, skills and attitudes or dispositions [22] which can be measured. Kavathatzopoulos [23] submits that just like other forms of competence which enable professionals to perform technical tasks, ethical competence can be acquired through education or training in ethical content, decision-making and problem-solving in ethics. However, that requires a psychological approach to ethics skills training, as opposed to the conventional pumping of theoretical ethics content [19].

The psychological approach involves exposing individuals to interactions with real life situations as part of their learning. This approach helps individuals to attain the skills and functional ability needed to process ethical problems by identifying the applicable rules to solve the problem without being constrained to previous ideas. When an individual attains that level of operation, such an individual is elevated to a level of competence which demonstrates ethical autonomy [23]. The success of this form of training for ethical competence is reported in a study conducted by [19]. The study cites a series of dedicated skills training interventions including the use of technology in the successful development of ethical competence.

### 2.2. Ethical Responsibility in Software Development

The next question is what is responsibility or what does it mean to be responsible? Although responsibility can be confused with accountability, Solbrekke and Englund [2] define responsibility as "*a moral obligation assumed by oneself, or bestowed upon a person to be used... to be and act for another*". Therefore, responsibility is the duty entrusted to a

professional by others who are served by the profession. Moral responsibility is not limited to an individual, but tends to be shared, as it extends accountability between the group and to its members. For software development, it also embraces the persons for whom the software is developed [7,15]. To the software engineer, it becomes a microlevel of moral duty assumed, which is for the provision of a specialised technical skill to advance the public good and serve others by developing software artefacts. Furthermore, the software engineering profession incorporates several societies to which software engineers belong, and software engineers work as a collective within these organisations. These forms of organisation create shared responsibility.

In an attempt to delineate the concepts of responsibility and accountability, [2] identifies several of the concepts' logical implications. In the interpretation of the contrasts between the two, it becomes apparent that responsibility emanates from a proactive self-volition instilled by the profession in whomever is entrusted with the moral duty, whilst accountability is inclined to reactive and rudimentary compliance. Professional responsibility embraces accountability, as it elevates a professional to the greater degrees of autonomy required to exercise professional judgement [2]. Being responsible requires prior commitment to performing good quality work and being accountable for consequences [10]. Therefore, the focus of responsibility is on the utilisation of specialist skills to competently evaluate alternative courses of action with regards to their costs and benefits (not necessarily financial but also ethical values) and select the one which advocates for the public good. In addition, it is also important to account for the outcomes of one's decisions and actions. This is exactly where ethical responsibility features. Of course, besides following the rules of good practice, the engineer should also object to decisions which are not in line with professional standards [24]. In the consideration of professional responsibility as an integral quality of a specialist which enables conscientious work performance, it provides a guarantee of acceptable levels of quality work even under challenging circumstances [1].

Furthermore, professional responsibility is based on mutual trust and respect between the software engineer, their profession and the public, and also on the professional's privileged status in society [25]. It is by the possession of specialist skills that this mutual trust and respect exists. Professional responsibility embraces the obligations on a profession and goes beyond personal obligations such as honesty and fairness [26]. In this case, the software engineer needs to demonstrate technical competence and moral reasoning [2] in a quest to contribute to ethically responsible software engineering [27]. Therefore, being ethically responsible should not be limited to one side of the coin, which is accepting responsibility after a problem has occurred, but also, in trying to prevent problems that lead to software catastrophes, software engineers take ethical responsibility throughout the software development process to ensure that all decisions and artefacts produced speak to the public good. This is what Gotterbarn [10] considers positive responsibility, as the engineer takes responsibility for the consequences of their work, as opposed to negative responsibility, in which the engineer seeks ways to absolve themselves of responsibility or disassociate themselves from a problem. Examples of negative responsibilities are outlined in the next paragraph. It is in respect of positive responsibility that trust between software engineers and whomever they serve can be established. In line with the prescripts of codes of ethics to promote the profession for the good of society, and for practitioners to enjoy the trust of others, they are expected to demonstrate capability in moral reasoning and meet the behavioural expectations on their profession in serving the needs of the public [2,10]. In this way the software engineer commits to a higher degree of care for society, which is directly and indirectly affected by the software artefacts [10].

One of the defining characteristics of a profession is the provision of a professional code of ethics which spells out the ethical standards which practitioners should adhere to. These codes also outline the moral responsibilities of the profession [7]. In analysing some of the popular codes of ethics' stances on the responsibilities of software engineers, Herkert et al. [28] defines the engineer's paramount responsibility as "*to protect the "safety, health, and welfare" of the public*" whilst Bittner and Hornecker [12] highlight the protection of "*public*

*interest*". In the software engineering area, for simplicity's sake, the IEEE-CS/ACM code of ethics [29] lists the ethical obligations in a catalogue of eight principles (i.e., public, client, product, judgement, management, profession, colleagues and self), with major emphasis on advocacy for the "*public interest*". Furthermore, this allows for the organisation of responsibilities according to the roles allocated to stakeholders in software development to whom the engineer is responsible [10]. Software engineers are expected to internalise these codes, which enlighten them regarding the ethical responsibilities entrusted to them by the profession they represent. The internalisation of the codes should be exhibited through ethical behaviour and competent engagement with ethical concerns which arise when dealing with the various activities involved in software development. However, some researchers have claimed that some software engineers have not "*smelled the coffee yet*", as they still do not take responsibility, and instead continue to defer the blame to others. For example, to evade responsibility, some engineers say that there are "*too many people*" and "*too many decisions*", according to Herkert et al. [27], that "*it's a bug*" or "*computer error*", according to Gotterbarn [10], and "*I am just an engineer*", according to Trim [30]. These are good examples of negative responsibility, as characterised by [10]. All these examples amount to the shifting of the blame to the computer itself [31] and issues around the development environment, instead of acknowledging the human fallibility of software engineers.

In summary, the form of responsibility provided and detailed above is the one chosen to guide this study. The summary concludes that software engineers are expected to take professional responsibility, specifically ethical responsibility for their work. This requires that they are aware of their ethical responsibilities [17]. This includes being aware and considering the ethical needs of stakeholders in software development. Furthermore, they also should have developed ethical competence which enables them to exercise ethical autonomy as well as complying to the set rules and obligations. This will enable them to be able to renounce unethical tendencies as part of taking ethical responsibility.

*2.3. Related Studies*

Continuous monitoring of professionals for their level of professional responsibility is important [1]. Hence, the subject of responsibility in software development has been investigated in the various research studies which we present below.

A study by Parnas [26] showed that software engineers should be aware of their responsibilities as professionals. This study determined that software engineers have personal, professional, and social responsibilities in their development of software. A study by Gotterbarn [10] examined how software developers avoided accepting responsibility for their work. The study showed that being a professional (as applicable to software engineers) assumes a broader or even higher level of responsibility, which includes the duty to observe ethical responsibilities spelled out in the professional codes of ethics. Both research papers indicate that the lack of understanding of what ethical responsibility is raises challenges for assuming such responsibility. Furthermore, the studies provided an expanded view of ethical responsibility and illustrated how professional and ethical responsibility should reside with the software engineers within software development. This expanded view is useful for guiding empirical studies on ethical responsibility in software development.

Paradice [32] conducted a survey study to determine the ethical attitudes of entry level computing software professionals. Amongst others, some of the findings of the study showed that software developers should be held responsible for the correctness of their work, and therefore had a professional responsibility in this area. The study compared the perceptions of two groups of undergraduates, with the results showing that there is a significant difference between the perceptions of computing and noncomputing undergraduate students about the ethical responsibilities of software practitioners. However, the study did not include professionals with experience practising as software engineers, nor did it compare the perceptions of practising engineers to those of job entry level undergraduate

students. A study conducted by [17], highlighted in the Section 1 of this paper which the current paper is expanding on, focused on the perceptions of computing academics about the ethical responsibilities of key stakeholders in software development. The findings of the study show that various software development key stakeholders should be held ethically responsible according to their contributions to software development. For example, developers, managers, and originators of software products were perceived to have high-level responsibility for ensuring the security of data, and in cases where software was used to do something unethical. Furthermore, the same study showed that software engineers should always consider the ethical implications of their software and be held responsible for the quality of their work. Although the study used a survey design for data collection, as in the current study, the data analysed only included responses collected from teachers of software development courses and excluded other key stakeholders such as industry-based practitioners, end-users and students studying software development courses amongst others. Although the studies by [17,31] focused on ethical responsibility, the latter focused more on attitudes towards assuming ethical responsibility whilst the former focused more on the levels of ethical responsibility on various areas within software development.

Before that, a study by Solbreke and Englund [2] which focused on the state of the formation of professional responsibility of technical specialist students was carried out. The study shows that, worryingly, the significance of moral aspects of professional responsibility seems to have been eroded and calls for the reshaping of professional responsibility by allowing a moral and societal mandate to be the driving force behind professional practice. Furthermore, the study shows the pitfalls of how accountability can cloud responsibility in governance systems. Such pitfalls are likely to weaken ethics governance systems aimed at ensuring that key stakeholders are held ethically responsibility in software development.

The above cited studies focused on professional responsibility by probing various aspects of the profession such as management, communication, ethics relating to the social implications of software and levels of ethical responsibility in software development. However, the cited studies have not juxtaposed the perceptions of students, practitioners, and educators in one study. We find it relevant in this study to juxtapose the responses of the three groups of respondents to determine if there are significant differences or not in the way they perceive the ethical responsibilities of software development key stakeholders in software development.

## 3. Methodology

The purpose of this study is to determine if there are differences in the perceptions of students, lecturers and software engineers employed in the industry regarding the ethical responsibilities attributed to various software development key stakeholders. To achieve the objective of this study data were collected through a survey which was shared on an online platform for 12 months from 2020 to 2021. A total of 561 responses were received, including 44 from computing academics, 103 from an unknown population of industry software practitioners and 414 student responses from a population of approximately 6000 students.

To ensure the validity of the questions in the survey, senior and experienced colleagues reviewed the questions. The data collection instrument was also reviewed through relevant research committees such as the ethics review committee within the university. Based on their feedback, the questions were improved. Furthermore, statistical tests to assess the validity of the data were conducted. Quantitative statistical techniques were used to analyse the data to ensure the results were as intended. A $p$-value of 0.05 was applied and if the calculated $p$-value was less than 0.05 then the study would conclude that there was a significant difference between what was being tested. The study employed a quantitative approach, using descriptive and variance statistical analysis to analyse the data.

The data contain responses to twelve questions coded as Resp1 to Resp12 as in Table 1. Resp# refers to a question in the data collection instrument used to probe for responses about a particular responsibility. The data relate to the ethical responsibilities of various

software development key stakeholders involved in software development regarding quality of work, testing of software, security of data processed by software and the use of software for illegal or unethical purposes.

**Table 1.** Responsibility statements.

| Responsibility Code | Category | Survey Question | |
|---|---|---|---|
| Resp1 | General Software Quality | Software developers should be held accountable for the quality of their work. | |
| Resp2 | Testing | Different people are involved in testing new systems or system changes before they go live. How would you rate the level of responsibility of the following people for testing systems? | [Programmers and developers] |
| Resp3 | | | [Business Analysts] |
| Resp4 | | | [Users] |
| Resp5 | | | [Project team] |
| Resp6 | | | [QA team] |
| Resp7 | Security of data | Many systems store personal data. Who do you believe is most responsible for the security of this data? | [The people who use the system] |
| Resp8 | | | [The programmers who developed the system] |
| Resp9 | | | [The company that owns the system.] |
| Resp10 | Illegal or unethical software | Sometimes software is used for an illegal or unethical purpose. (For example: many VW cars sold in America had software that could detect when they were being tested and change the performance accordingly to improve results.) Who do you believe is responsible when software does something unethical? | [The people who proposed the idea or design.] |
| Resp11 | | | [[The developers who created the software.] |
| Resp12 | | | [Management who approved and/or paid for the software.] |

To assess the responses, descriptive statistics in the form of frequencies and percentages were used to summarise the responses, whilst the mean and standard deviation were applied to evaluate the responses (that is [Very responsible], [Partly responsible], [Not really responsible] and [Do not know]).

## 4. Research Results and Discussion

This section provides the research results of this study. We first present the demographic analysis of the respondents, followed by the descriptive statistics of the responses. Conclusions, contributions and implications of the findings are presented before a discussion on limitations and recommendations for future studies are presented.

### 4.1. Respondents Demographics

The research results in Table 2 show that 561 (44 from computing academics, 103 from industry software practitioners (corporate) and 414 students) useful responses were received to be analysed in order to answer the research question of the study. Useful responses for this study were all questionnaires which were completed without missing values. Females constituted 161 (28.70%) of the respondents and males 371 (66.13%). Amongst the respondents, 29 (5.17%) preferred not to state their gender.

**Table 2.** Demographic information of the respondents of the study.

| | Lecturers | | | | Corporate | | | | Students | | |
|---|---|---|---|---|---|---|---|---|---|---|---|
| | Description | Freq | Percent | | Description | Freq | Percent | | Description | Freq | Percent |
| Gender | Female | 8 | 18.18 | Gender | Female | 12 | 11.65 | Gender | Female | 141 | 34.06 |
| | Male | 34 | 77.27 | | Male | 86 | 83.50 | | Male | 251 | 60.63 |
| | #1 | 2 | 4.55 | | #1 | 2 | 3.88 | | #1 | 20 | 4.83 |
| | | | | | Other | 1 | 0.97 | | Other | 2 | 0.48 |
| | Total | 44 | 100 | | Total | 103 | 100 | | Total | 414 | 100 |
| Age group | 30–39 | 30 | 68.18 | Age group | 18–29 | 30 | 29.13 | Age group | 18–29 | 402 | 97.10 |
| | 40–49 | 8 | 18.18 | | 30–39 | 33 | 32.04 | | 30–39 | 10 | 2.42 |
| | 50–59 | 6 | 13.64 | | 40–49 | 25 | 24.27 | | 40–49 | 2 | 0.48 |
| | | | | | 50–59 | 13 | 12.62 | | | | |
| | | | | | 60 and older | 2 | 1.94 | | | | |
| | Total | 44 | 100 | | Total | 103 | 100 | | Total | 414 | 100 |
| Highest qualification | Degree | 8 | 18.18 | Highest qualification | Degree | 29 | 28.16 | Highest qualification | #5 | 1 | 0.24 |
| | Diploma | 1 | 2.27 | | Diploma | 32 | 31.07 | | #6 | 1 | 0.24 |
| | Doctorate | 1 | 2.27 | | Matric | 10 | 9.71 | | Bachelor | 1 | 0.24 |
| | Post-grad | 33 | 75 | | #2 | 5 | 4.85 | | Degree | 4 | 0.97 |
| | #1 | 1 | 2.27 | | #3 | 1 | 0.97 | | Diploma | 88 | 21.26 |
| | | | | | #4 | 25 | 24.27 | | #7 | 1 | 0.24 |
| | | | | | Other | 1 | 0.97 | | #8 | 1 | 0.24 |
| | | | | | | | | | #9 | 1 | 0.24 |
| | | | | | | | | | #10 | 1 | 0.24 |
| | | | | | | | | | Matric | 301 | 72.71 |
| | | | | | | | | | #11 | 13 | 3.14 |
| | | | | | | | | | #12 | 1 | 0.24 |
| | Total | 44 | 100 | | Total | 103 | 100 | | Total | 414 | 100 |
| Dev exp. | 0–5 years | 18 | 40.19 | Dev exp. | 0–5 years | 27 | 26.21 | | | | |
| | 6–10 years | 6 | 13.64 | | 6–10 years | 21 | 20.39 | | | | |
| | <1 | 1 | 2.27 | | <1 | 55 | 53.40 | | | | |
| | >10 | 13 | 29.55 | | >10 | 0 | | | | | |
| | None | 6 | 13.64 | | None | 0 | | | | | |
| | Total | 44 | 100 | | Total | 103 | 100 | | | | |
| Teaching exp. | 1–2 years | 7 | 15.91 | | | | | SYC | 1st year | 119 | 28.74 |
| | 3–5 years | 12 | 27.27 | | | | | | #14 | 24 | 5.80 |
| | 6–10 years | 15 | 34.09 | | | | | | #15 | 41 | 9.90 |
| | Less than 1 | 1 | 2.27 | | | | | | 2nd year | 153 | 36.96 |
| | >10 | 8 | 18.18 | | | | | | 3rd year | 77 | 18.60 |
| | None | 1 | 2.27 | | | | | | Total | 414 | 100 |
| | Total | 44 | 100 | | | | | | | | |

#1—Prefer not to say. #2—Did not complete matric/grade 12 (i.e., a qualification obtained at the end of secondary schooling). #3—Partial masters degree. #4—Post-graduate qualification. #5—2nd year student. #6—4th year student. #7—Final year diploma student. #8—Doing my second year in national diploma in software. #9—Higher certificate. #10—N6 certificate. #11—Post-graduate qualification. #12—Still studying towards my diploma in IT. #14—Fourth year or later. #15—None: I am a first year student. Dev exp.—Development experience. Teaching exp.—Teaching experience. SYC—Study year completed.

The corporate and lecturer groups comprised respondents in the age range of 30–39 in higher percentages compared to the other age groups. On the other hand, the student group had 402 (97.10%) respondents in the age range of 18–29. Furthermore, 301 (72.71%) of the students had only matric as highest qualification, while only 107 (25.85%) respondents in this group had post-matric qualifications. On the other hand, 34 (77.27%) lecturers had post-graduate qualifications, whereas 26 (59.09%) industry-based practitioners possessed postgraduate qualifications. Just more than half (53.4%) of the total respondents had more than 10 years of working experience, while 20.39% reported to have 6 to 10 years of working experience. The remaining 26.21% of the participants had 0 to 5 years of working experience. According to the results, software development experience for lecturers is as follows: 29.55% had more than 10 years of experience; 13.64% had between 6 and 10 years of experience; and 40.91% possessed between 1 and 5 years of software development experience. The majority of the lecturers (61.36%) had combined teaching experience of between 3 and 10 years.

*4.2. Respondents' Perceptions of Ethical Responsibilities*

This section presents the results of the survey pertaining to the ethical responsibility questions posed to the respondents.

4.2.1. Responses of Individual Categories: Students, Corporate Practitioners and Lecturers

Table 3 presents the frequencies and percentages of the responses for each of the three categories of respondents for each group of responses to the ethical responsibility questions on the Likert scale. The table summarizes the total responses for the 12 items, represented as responsibility statements in Table 1, to provide frequencies and percentages, per group, of responses for each category of respondent. For all three categories, the "*Very responsible*" option was selected by the majority (students (63%), software practitioners (57%), lecturers (72.35%)) of the respondents followed by "*Partly responsible*". On the other hand, "*Not really responsible*" was the second least selected option, after the "*Do not know*" option, by respondents from the three categories.

**Table 3.** Summary of response results for each category of respondents.

| | Students | | Corporate Practitioners | | Lecturers | |
|---|---|---|---|---|---|---|
| | **Freq** | **Percent** | **Freq** | **Percent** | **Freq** | **Percent** |
| Very responsible | 261 | 63.08 | 58 | 57.85 | 32 | 72.35 |
| Partly responsible | 112 | 27.09 | 29 | 28.16 | 9 | 20.45 |
| Not really responsible | 35 | 8.51 | 13 | 12.30 | 3 | 7.20 |
| Do not know/Don't have this | 6 | 1.31 | 2 | 1.70 | 0 | 0 |
| Total | 414 | 100 | 102 | 100 | 44 | 100 |

4.2.2. Are There Significant Differences amongst the Groups of Responses for the Three Categories of Respondents?

To answer the question of whether there are significant differences in the way the three categories of respondents answered the survey questions in terms of the four groups of responses on the Likert scale, in Table 4 we present sums, averages and variances in terms of the responses from the three categories combined. Whilst count refers to the categories of respondents, the "Sum %" represents the total sum of the percentages of responses for each group ("Not really responsible", "Partly responsible", "Very responsible", and "Do not know/Don't have this") across the different categories (Student, Corporate, Lecturers). The sum is calculated to provide an overall perspective on the distribution of responses for each responsibility level across the different categories. The "Average %" in this context represents the average percentage of responses for each group ("Not really responsible",

"Partly responsible", "Very responsible", and "Do not know/Don't have this") across the different categories (Student, Corporate, Lecturers). It is the average of the percentages reported for each group in the respective categories. These average percentages are calculated to provide an overall summary of the distribution of responses for each responsibility level across the different categories. SS refers to Sum of squares, DF to degrees of freedom, MS represents mean of square, and F refers to F value.

**Table 4.** The average and variances of the responses for the three respondent categories to the ethical responsibility questions combined.

| Groups of Responses | Count | Sum % | Average % | Variance % |
|---|---|---|---|---|
| Very responsible | 3 | 192.191 | 64.06 | 61.47996 |
| Partly responsible | 3 | 75.70328 | 25.23 | 17.41739 |
| Not really responsible | 3 | 28.00455 | 9.33 | 7.014923 |
| Do not know/Don't have this | 3 | 2.931344 | 0.98 | 0.726604 |

The mean values (average %) show that the "*Very responsible*" option was the most selected choice (64.06%), followed by "*Partly responsible*" (25.23%), while "*Not really responsible*" was the second least (9.33%) and "*Do not know*" was the least (0.98%) selected option. Furthermore, the results presented in Table 5 also indicate that there was a significant difference (since *p*-value < 0.05) in the way the respondents from the three categories combined answered the questions related to ethical responsibility.

**Table 5.** ANOVA Test.

| Source of Variance | SS | Df | MS | F | *p*-Value | F-Crit |
|---|---|---|---|---|---|---|
| Between groups | 7045.448466 | 3 | 2348.483 | 108.4263 | 0.00001< | 4.066181 |
| Within groups | 173.2777422 | 8 | 21.65972 | | | |
| Total | 7218.726208 | 11 | | | | |

In summary, the results presented in Sections 4.2.1 and 4.2.2 show that the majority of the participants believe that software development key stakeholders should be held ethically responsible for their respective roles. This perception prevails across the three groups of respondents (students, lecturers and software practitioners) for all the ethical responsibility questions posed to the respondents, as may be seen from Table 3 (individual groups), Table 4 (the three groups as a collective) and Table 5 (ANOVA test).

The fact that the respondents believe that software development key stakeholders should be held ethically responsible for the quality of their work is encouraging, as it signals an understanding by the practitioners about the responsibility entrusted in them by society.

Furthermore, software engineers are involved in the testing of software to ensure it complies with its requirements. Although the testing is conducted at different levels by various people responsible for software quality, this study's results show that all software development key stakeholders are perceived to be ethically responsible for ensuring that software is tested for quality.

In terms of the security of data, software packages process lots of data which require protection, especially data including confidential information about people. Once more, the study shows that the common view amongst the respondents is that all software development key stakeholders should assume ethical responsibility for the way data are collected and used by software systems. Lastly, in the same vein as with the security of data, the responses show that all key stakeholders involved in the creation of software are responsible for ensuring that software is not used for illegal purposes, but rather that its development and use should be for ethical purposes.

#### 4.2.3. Is there Any Statistical Significance in the Differences in the Ethical Responsibility Perceptions of the Three Respondent Categories?

Table 6 presents the averages and variances of the responses by students, lecturers and industry-based practitioners on their perceptions regarding the ethical responsibilities in software development, as laid out in Table 1. In this case count refers to the number of groups of responses to the Likert scale used in the survey. Sum refers to the total percentages for all the groups of responses per category of respondents. The individual average values (25% for students, 24.7% for software practitioners and 25% for lecturers) of the responses of the three groups are very similar. The similarity in responses of the three groups is also confirmed by the ANOVA test results shown in Table 7, where the *p*-value of the F-test is 0.999846, which is greater than 0.05, thus indicating that there is no statistically significant difference between students, corporate and lecturers' responses.

**Table 6.** Averages and variances of the three respondent categories.

| Categories of Respondents | Count | Sum | Average | Variance |
|---|---|---|---|---|
| Students | 4 | 100 | 25.0 | 762.6315 |
| Corporate | 4 | 100 | 25.0 | 576.1982 |
| Lecturers | 4 | 100 | 25.0 | 1068.163 |

**Table 7.** ANOVA Test.

| Source of Variance | SS | Df | MS | F | *p*-Value | F-Crit |
|---|---|---|---|---|---|---|
| Between groups | 0.24738 | 2 | 0.12369 | 0.000154 | 0.999846> | 4.256495 |
| Within groups | 7220.977 | 9 | 802.3308 | | | |
| Total | 7221.224 | 11 | | | | |

The similarity in responses of the three categories shows that there is no significant statistical difference in how students, educators and software practitioners perceive the ethical responsibilities of software development key stakeholders. Therefore, this provides an answer to the study's research question. This shows that software engineers across the spectrum, whether they are at the entry level of the profession, contributing to the teaching of software engineering, or practising in the development of software, they share the same views regarding the ethical responsibilities of key stakeholders. Nevertheless, the challenges we are faced with at the advent of emergent technologies is the continued violation of ethics despite this awareness. This may suggest a need for a transformation in terms of how software engineers should be held responsible for ethical violations. Shakib and Layton [33] also suggest that the failure of self-regulation to address ethical violations in the software industry will force the legal system to enact more laws.

#### 5. Conclusions, Contributions and Implications

This study sought to compare the perceptions of software development engineers, namely students, lecturers and software practitioners, by asking them about their perceptions of the ethical responsibilities of software development key stakeholders. Firstly, the results show that the majority of the respondents across the three categories believe that software practitioners are to be held ethically responsible for their behaviour. This view should bring comfort to millions of consumers of software products. However, judging by numerous reports of the incidences of unethical behaviour from some software practitioners, the expressed perception is cold comfort at the same time.

The study found that there is no significant statistical difference in how students, educators and software practitioners perceive the ethical responsibilities of software development key stakeholders. Simply put, all the three categories believe that key stakeholders in software-related activities must be held ethically responsible for their actions. This

finding is a key contribution of our study to the field of software engineering ethics. Many of the prior studies on ethical responsibility mainly focused on a particular group (either students, academics or those in practice) of software engineers. For example, studies by [2,31] focused on students, whilst a study by [17] focused on software engineers in academia. This study extended the inquiry into ethical responsibility to include those who are in the software development industry. The insights brought by this extension are a further contribution to the body of knowledge.

Lastly, the organisation of the responsibilities related to key stakeholders in software development, as shown in Table 1, provides a framework which can be used by other researchers pursuing studies in this area. The development of this framework is yet another contribution of the study to the body of knowledge.

### 6. Limitations and Future Studies

This study, as is the case with other research studies, has limitations which should be considered when reading it. Firstly, the study's data were collected within the South African environment and the study did not include all key stakeholders in software development. Therefore, the findings cannot be generalized to software development environments in other countries, as their cultures, practices and experiences may be different. Extending this study to other countries, and also including other key stakeholders such as users of software, can benefit the understanding of how key stakeholders view the ethical responsibilities of software engineers. This will further assist to remedy the limited scope of this study in terms it being localized to one country and having not included an extensive list of key stakeholders.

Furthermore, future studies can include an examination of the demographic elements of the respondents in relation to how they view the ethical responsibilities of software engineers. Conducting such studies can assist in generalizing findings across the relevant environments and demographic elements of the respondents.

**Author Contributions:** Conceptualisation, S.M.M. and R.T.H.; Literature review, S.M.M.; Methodology, S.M.M.; Analysis, S.M.M. and R.T.H.; Results reporting, S.M.M. and R.T.H.; writing the original draft, S.M.M.; writing—review & editing, S.M.M. and R.T.H. All authors have read and agreed to the published version of the manuscript.

**Funding:** The APC was funded by Tshwane University of Technology (TUT).

**Institutional Review Board Statement:** The study was conducted in accordance with ethics approvals from TUT's Faculty Committee of Research Ethics, with reference number FCRE/ICT/2019/07/001 (ICT), and the Research Ethics Committee, with reference number REC2019/10/004.

**Informed Consent Statement:** Informed consent was obtained from all subjects involved in the study.

**Data Availability Statement:** The data presented in this study are available on request from the corresponding author.

**Acknowledgments:** The authors would like to acknowledge the following people: Jacqui Coosner for her contribution to the data collection and Livhu Nedzingahe for supporting with statistical data analysis.

**Conflicts of Interest:** The authors declare no conflict of interest.

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
