# Peer review of "A Comparative Study on the Ethical Responsibilities of Key Role Players in Software Development"

_2674-113X, doi:10.3390/software2040023_

Round 1
Reviewer 1 Report
Comments and Suggestions for Authors
The paper is very interesting. The research is well conducted and the results are very well explained, The paper has a real value to contribute. However, some changes are required:
a) The introduction should be improved. For instance, hypothesis formulations should happen in a separate section, which should be supported by literature, If you prefer to do this in the introduction then make sure to include all major references without much explanation (as they will be explained in related work).
b) Related work needs to be improved as gaps are not very well explained.
c) Research methodology is good but include statements about how you ensured validity.
d) Explain the results and provide a rich discussion that goes beyond what you planned to achieve (applications of your work)
Comments on the Quality of English LanguageSome grammatical mistakes are there. Kindly correct.
Author Response
Good day
Kindly note that our responses to the reviewer's comments are in the attached file.
Regards

Reviewer 2 Report
Comments and Suggestions for Authors
Summary: This paper presents an empirical study that examines how ethical responsibility is perceived among different software development roles (software designers, developers, managers) by various groups (students, teachers, practitioners). The study is interesting and well-written, but there are some concerns, including issues with survey design and data presentation.
Survey Design and Data Presentation: Tables 4-7 present some challenges. The meaning of "count" should be clarified. Also, it is unclear how the high values for “sum”, “average”, etc. have arisen. Additionally, the definitions of "sum," "average," and "variance" need explanation to improve understanding. These issues also raise doubts about the validity of the ANOVA test results. Since Section 4.2.2 and Section 4.2.3 lead to the paper's main findings, it is crucial to address and rectify these issues to ensure the paper's acceptability for publication.
Terminology Clarification: The term "role-players" is ambiguous (reminiscent of role-playing game players) and should be replaced with more precise terminology, such as "software development key stakeholders" or something similar. Consistency in terminology usage is also needed throughout the paper.
Definition of "Responsibility": The paper should provide its own concise definition of "responsibility" in the introduction and refer to the section that later discusses how it can be defined. This will provide clarity and context for readers.
Reference to Previous Study: The introduction mentions that this study is an extension of a previous study. It is essential to provide a reference to the previous study if it is published (is this ref11?) and indicate that it will be discussed later in the paper. Furthermore, the later discussion of the previous study should be highlighted in the related work section, emphasizing the differences in methodology and scope. Also, consider aligning the descriptions in introduction and related work, e.g., the inclusion or exclusion of quality assurance professionals and end-users.
Full Survey Questionnaire: To enhance transparency and reproducibility, the paper should include the full survey questionnaire, including demographic items and other relevant questions. Additionally, it should confirm whether all presented items had response options of "very responsible," "partly responsible," "not really responsible," and "do not know."
Correlation Coefficients for Demographics: Consider using correlation coefficients to explore relationships between respondents' demographics. This can help highlight similarities or differences among different groups, enhancing the paper's analysis.
Clarification of Specific Terms: The term "Matric" is specific to South Africa and should be explained for readers who may not be familiar with it.
Use of "Useful" Responses: The paper mentions 561 "useful" responses, but it's unclear how these were determined. The criteria for classifying responses as "useful" should be explained to ensure transparency.
Table 2 and Data Presentation: In Table 2, the use of "#" without an explanation should be avoided. Abbreviations could be used in the table instead of numbering items. Experience in years should be ordered logically to avoid confusion. Using "<1" and ">10 years" is a suggestion to avoid some “#”-numbers. Consistency in row headings within the "dev. exp." block is essential for clarity.
Data Availability Statement: While the authors state that "the research data used in this study is available," the paper should explain how the data can be accessed. Providing the data as supplemental material, as allowed by the journal, would be beneficial, especially if the paper aims to serve as a "launch pad" for further studies.
Questionnaire Design: In subsequent studies, consider using Likert scale items for questionnaire questions to allow for more robust statistical analysis and reduce potential misinterpretations of agreement levels. Additionally, Resp12 may be too simplistic, as managers often have both financial responsibilities and disciplinary power, which should be accounted for in the questionnaire. Of course, this cannot be fixed in this paper.
In conclusion, the paper presents an intriguing study but has several issues that need to be addressed, particularly in survey design, data presentation, terminology clarification, and data availability. Resolving these concerns will significantly improve the paper's quality and its suitability for publication.
Comments on the Quality of English LanguageOnly some minor issues. Language is mostly fine.
Author Response
Good day
Kindly note that our responses to the reviewer's comments are in the attached file.
The survey questionnaire is available, we can email it as we are only allowed to attach one file here.
Regards

Reviewer 3 Report
Comments and Suggestions for Authors
The references in the introduction section could be enhanced. For example, the paper mentioned that "software developers are inevitably confronted with the constant need to make decisions based on competing needs such as client demands, cost, operational efficiencies ...". This statement is true, but it would be better to cite related materials to support your statement, such as what's the wrong decision made by the developer when they are rushing to develop a client demand or other requirements, such as the existing work (DroidPerf: Profiling Memory Objects on Android Devices [MobiCom'23]) shows a lot of bad coding styles/problems due to the limited development timeframes in mobile app field.
Same to other statements in the introduction, such as the increased society's dependence on technology increases its vulnerability through technological failures (give a support example here?).
Could you give a better motivation of why the paper needs the two hypotheses, does it make the focused research problem harder or easier to study?
In section 4.1, it's good to see a detailed description of Table 2 (where the data comes from and how it's distributed across different factors, such as gender, age group, and highest qualification). But could you give a short summary of your description? Like, what does the data indicate?
The future studies look good to me, but the limitations should be expanded more.
Comments on the Quality of English LanguageNit: Please check the punctuation usage, for example, missing period after citation [7].
Author Response

(The authors gave the same response as above.)

Round 2
Reviewer 1 Report
Comments and Suggestions for Authors
The authors have incorporated almost all my comments. Regarding the implications, the findings could have implications in different aspects, for instance, AI technology providers and technology adopters. These days people are talking about AI-driven software engineering. So, who assumes the ethical role-technology providers or adopters? Its a kind of shift in responsibility?
This for sure not explicitly dealt with in the paper, but various such aspects can be included, helping readers find future research directions.
Anyways, the paper in its current form is acceptable.